# Non-random patterns in the co-occurrence and accumulation of adverse life events in two national panel datasets
Kyra Evers [1]✉, Denny Borsboom[1], Eiko Fried[2], Fred Hasselman [3], František Bartoš[1] & Lourens Waldorp [1]

Adverse life events (ALEs), such as illness, bereavement, and accidents, can have profound consequences for physical and mental health. Although existing research highlights risk factors for ALEs, such as personality and socioeconomic status, less is known about patterns in ALEs themselves. How do events cluster and accumulate over time? Using generalized linear mixed-effects models, we study yearly self-reported ALEs in two panel datasets, the Swiss Household Panel (n = 16,946, 210,031 person-years) and the Household, Income and Labour Dynamics in Australia (n = 25,803, 113,605 person-years). We identify widespread contemporaneous and lag-1 associations between ALEs. The twenty-year accumulation of ALE counts deviates substantially from a random process and is better described by a self-reinforcing process, in which ALEs increase the risk of future ALEs. For all analyses, differences between individuals and households were stronger predictors of event occurrence than concurrent or prior adverse life events. Non-random patterns in ALEs should inform our conceptual and statistical models, as well as our prevention strategies.

Adverse life events happen to virtually everyone. A family member's illness, a car accident, job loss, or a close friend's death – these events touch every person at some point in their lives. Although around two-thirds of people show resilience in the face of adversity[1], and some adversity may be beneficial[2], adverse life events can have profound detrimental consequences. Higher counts of adverse life events have been associated not only with lower well-being and life dissatisfaction[3], but also higher rates of stress-related outcomes, such as physical illness (e.g., cardiovascular disease and infectious diseases[4]), functional impairment[5], mental illness (e.g., depression and anxiety[6–11]), and even death[12,13]. These outcomes are thought to be mediated by mechanisms such as disrupted affect regulation, engagement in unhealthy behaviours, and altered neuro-hormonal and autonomic nervous system functioning[14]. The effects of adverse life events can extend across multiple domains of life[14], potentially affecting not only physical and mental well-being, but also developmental, financial, occupational, and social health. As such, adverse life events not only affect individuals, but builds up to severe collective and societal consequences[15]. Understanding the nature and patterns of these events is therefore critical for identifying how they influence resilience, vulnerability, and opportunities for intervention and prevention. The present study investigates statistical patterns in how events co-occur and accumulate, exploring to what extent these events are non-random.

Multiple lines of evidence indicate that adverse life events do not simply "fall randomly from the sky"[14]: there are structural patterns in *who* gets *which* event. A classical distinction is made between dependent and independent life events[16]. Dependent life events are influenced by a person's characteristics and behaviour, including events such as relationship problems, unemployment due to underperformance, and financial issues due to overspending. In contrast, independent life events are considered as more "fateful" and beyond an individual's control[17], for example, a natural disaster or the death of a spouse. Though independent life events are typically seen as being more due to chance, structural patterns emerge for both dependent *and* independent life events. This challenges the notion that adversity is simply a matter of "bad luck"[9]. Rather, research in behavioural genetics, psychology, sociology, and stress and life course studies has identified key drivers of these structural patterns, including personality, psychopathological, and behavioural characteristics, socio-economic factors, and stress proliferation.

Personality, psychopathological, and behavioural characteristics are primary predictors of experiencing adverse life events. Twin studies indicate that variance in adverse life events in part reflects genetic variance, meaning that genetic factors partially influence an individual's exposure to adverse life events (i.e., gene-environment correlation[18–20]). Heritability estimates for dependent events are larger than those for independent events, with

[1]Psychological Methods, University of Amsterdam, Amsterdam, The Netherlands. [2]Department of Psychology, Leiden University, Leiden, The Netherlands. [3]Behavioural Science Institute, Radboud University, Nijmegen, The Netherlands. ✉e-mail: kyra.c.evers@gmail.com

estimates ranging from 31–45% and 7–18%, respectively[16,21–24]. Heritability appears to be mediated by personality and behavioural characteristics[25], as estimates shrink considerably or disappear completely when controlling for traits such as neuroticism, psychoticism, oppositionality, delinquency, physical aggression, depression, and anxiety[25–27]. Genome-wide association studies further suggest that common genetic variants associated with adverse life events overlap with those linked to neuroticism, major depressive disorder, and other stress-related traits[28]. This overlap supports the hypothesis that shared genetic factors may predispose individuals to both heightened adverse life event exposure and vulnerability to mental health challenges, potentially amplifying the impact of adversity through correlated pathways[28,29]. Importantly, evidence from genetically informed co-twin control designs indicates that, although associations between adverse life event exposure and mental health outcomes are attenuated within twin pairs, they nonetheless remain modest to considerable in magnitude, suggesting that these links cannot be fully explained by shared genetic or familial confounding[9,11].

In models that incorporate gene-environment correlations, adverse life events are not purely exogenous environmental stressors[27]. The mechanisms by which personality, psychopathological, and behavioural characteristics influence exposure to adverse life events include both causal and non-causal pathways[18]. Causal pathways involve passive, evocative, and active gene-environment correlations[18,30]. Passive gene-environment correlation occurs when parents transmit both genetic propensities and correlated environments to their offspring; evocative gene-environment correlation occurs when genetically influenced traits elicit reactions from others; and active gene-environment correlation occurs when individuals select or create environments aligned with their genetic propensities. Research from clinical and personality science elaborates on the mechanisms by which personality, psychopathological, and behavioural characteristics can contribute to dependent life events, a phenomenon termed *stress generation*[31–35]. The stress generation framework identifies four mechanisms[32,36,37], including *maintenance*: actively or passively staying in an environment, such as an unstable job; *evocation*: provoking certain reactions in the environment, such as relationship conflicts; *selection*: choosing to leave or enter an environment, such as marrying an aggressive partner; and *modification*: actively changing or creating a new environment, such as starting a risky business venture[32]. These mechanisms illustrate how genetically influenced personality traits and behaviours can increase exposure to dependent adverse life events through environmental shaping. Non-causal pathways in which exposure to adverse life events is influenced by personality, psychopathological, and behavioural traits include differences in framing, appraisal, recall, and reporting of events[18]. For instance, neurotic traits can lead to a more negative evaluation of events and increase the probability of endorsing adverse events in surveys[38]. This is supported by higher heritability estimates of self-reported compared to objectively rated events[22,39], meaning heritability estimates in part describe the subjective experience of events. Overall, the reciprocal interaction between the individual and their environment can establish and engrain structural patterns in adverse life events[40].

From a sociological perspective, structural patterns in who experiences adverse life events also arise due to differences in socio-economic status (SES). SES broadly includes educational attainment, occupation, income, and wealth[41]. SES stratifies risk of exposure to adverse life events, with lower SES often linked to correlated risk factors such as unsafe housing, residing in economically deprived and more criminal neighbourhoods, financial instability, and elevated exposure to environmental pollutants, toxins, and disaster-prone areas[42–47]. Accordingly, lower SES is associated with higher counts of adverse life events across domains[13,48–51], including more interpersonal events[42]. Moreover, structural differences in risk exposure can be amplified over time through vicious cycles (i.e., amplifying or positive feedback loops). In the social sciences, this is also known as cumulative disadvantage, which is "the systemic tendency for inter-individual divergence in a given characteristic (e.g., money, health, or status) with the passage of time"[41,52,53]. Terminology for this phenomenon varies across research domains, including the Matthew effect, preferential attachment[54], self-reinforcement[55], path-dependency, first-mover advantage, or colloquially as "the rich get richer and the poor get poorer"[56]. Over time, cumulative disadvantage can create large structural disparities in how adverse life events accumulate.

Finally, structural patterns in adverse life events can arise from the spreading effects of events *themselves*. In the stress and life course literature, this is known as *stress proliferation*[57], in which stressors can expand within the same domain, generate new stressors across life domains, and propagate stressors across people, such as households, families, and social groups. Stress proliferation mainly occurs because life domains such as work, family, and health are linked due to the multiplicity of a person's social roles ("linked lives"[58]): a person is not merely an employee, but also a parent, a child, a spouse, a friend, and so on. These life domains all contribute to the demands placed on a person, but also to the *resources* they have available, such as material wealth, social support, and personal characteristics such as coping strategies[59,60]. Resources both lower the risk of adverse life events and help cope with them if they do occur. Importantly, resources are not static, but may in turn be affected by adverse life events. On the one hand, experiencing adversity can help to build resilience to future adversity[2,59,61–63], also known as post-traumatic growth[64,65]. On the other hand, adverse life events have been linked to the onset of mental illness[9,11,66,67], diminished self-concept and self-esteem[68–70], and shattered core assumptions about safety and control[71,72]. Through such pathways, an adverse life event that depletes resources thus increases vulnerability and lowers resilience[59], such as a divorce, which leads to lower savings, a loss of friendships, and emotional dysregulation. This forms another positive feedback loop in which the occurrence of one adverse life event can increase the probability of a new adverse life event.

In summary, exposure to adverse life events is distinctly non-random, involving a complex web of numerous common causes and self-reinforcing pathways. Over time, their interaction has the potential to culminate in clusters of adversity and large individual differences. Whereas most research has focused on identifying risk and resilience factors of adverse life events or quantifying their impact, few studies examine patterns in events themselves[10,73]. Given the multivariate and interacting nature of these processes – personality, psychopathological, and behavioural characteristics, socio-economic factors, and stress proliferation – there are compelling reasons to expect structural patterns in how adverse life events cluster and accumulate. However, the specific patterns that emerge from these processes are not straightforward to predict. Which types of events tend to co-occur? Which events predict which subsequent events? How do adverse life events accumulate over decades?

The present study investigates statistical patterns in the co-occurrence and accumulation of adverse life events. Specifically, we use generalized linear mixed-effects models to examine contemporaneous (within-year) and lagged (1 year) associations between event types, autocorrelation in event counts, and deviations from randomness in twenty-year cumulative counts of adverse life events in two panel datasets: the Swiss Household Panel (SHP)[74,75] and Household, Income and Labour Dynamics in Australia (HILDA)[76,77]. To characterize the accumulation of events, we compare a "bad luck" model, in which risk is homogeneously distributed across the population, to two alternative stochastic processes: a frailty model, in which risk is heterogeneously distributed across individuals and households, and a self-reinforcing model (Polya urn), in which baseline risk is also heterogeneously distributed across the population, but events also increase the risk of future adverse life events. Examining structural patterns and self-reinforcing pathways in adverse life events informs our conceptualization of independent versus dependent adverse life events, as well as our design of prevention strategies to avert adversity cascades.

## Methods
### Data
The co-occurrence of adverse life events and their accumulation across the population can be studied using national survey studies, which

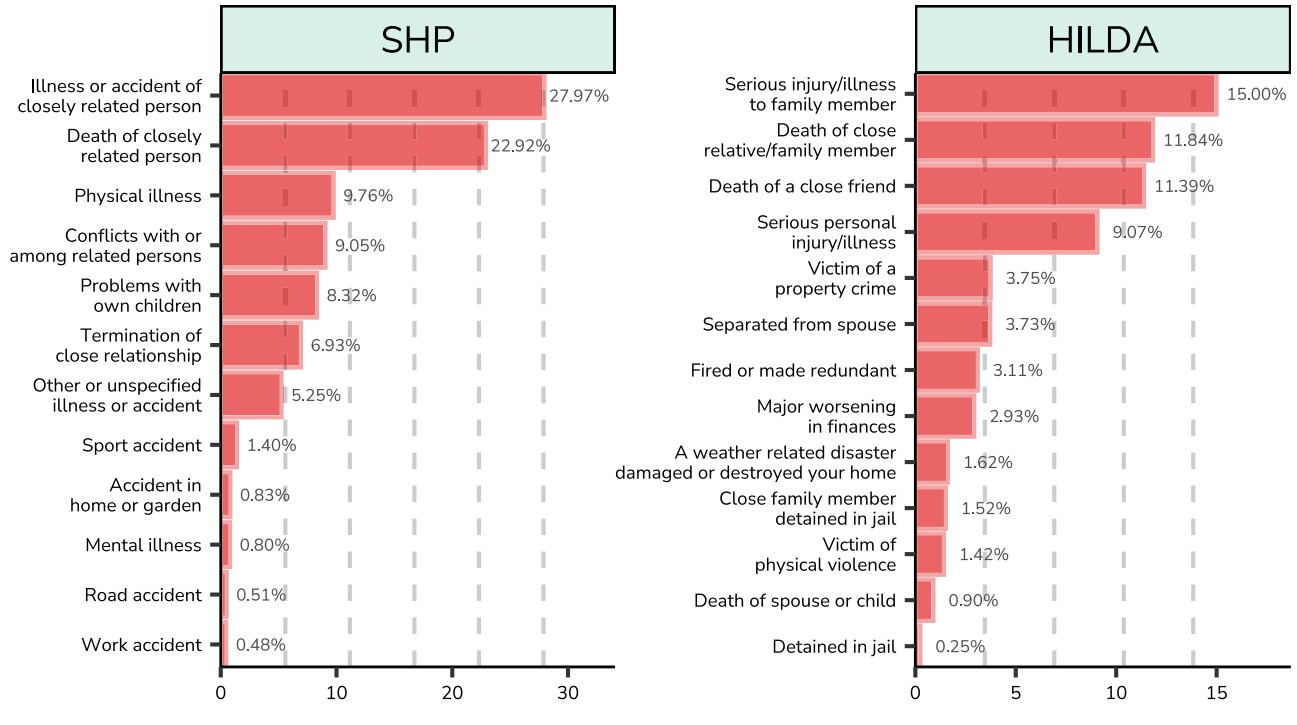

**Fig. 1 | Relative frequency of adverse life events across person-years.** The frequency is based on a sample of 27,298 individuals and 15,931 households with 179,310 total person-years (Source: Swiss Household Panel, SHP) and 30,637 individuals and 37,792 households with 289,083 total person-years (Source: Household, Income and Labour Dynamics in Australia, HILDA). Dashed lines indicate 10,000 instances.

longitudinally monitor a large, representative sample across decades. The present study analyzed two openly accessible survey datasets: the Swiss Household Panel (SHP)[74,75,78] and Household, Income and Labour Dynamics in Australia (HILDA)[76,77,79]. SHP and HILDA are large household panel surveys that broadly track the well-being of the Swiss and Australian population using self-report since 1999 and 2001, respectively. HILDA mainly collects data via face-to-face interviews, with a minority of phone interviews, while SHP mainly collects data via phone interviews, with a small minority of face-to-face interviews and a growing number of web-based collected data.

We included waves 2 (year 2000) until wave 24 (year 2022) for SHP and wave 2 (year 2002) until wave 23 (year 2023) for HILDA. Both SHP and HILDA include a module on life events since 2000 and 2002, respectively. SHP focuses more on interpersonal adverse events, whereas HILDA includes events from a wider range of domains[80]. We selected only closed questions that were asked consistently in each wave and that were considered as unambiguously adverse. We deliberately chose to focus on adverse rather than stressful life events, as even positive events such as getting married can be very stressful[40]. This yielded 12 (SHP) and 13 (HILDA) adverse life events (see Fig. 1). Events concerning a personal illness or accident in SHP were mutually exclusive, as only one such event was recorded per year. As such, the events physical illness, mental illness, work accident, road accident, accident in home or garden, sport accident, and unspecified illness or accident are mutually exclusive subcategories of a personal illness or accident, and could not co-occur.

Given the adult nature of many events (e.g., divorce, death of spouse, fired), we only included observations where individuals were 18 years and older, excluding 11,089 (SHP) and 15,033 (HILDA) person-years. Person-years for which all adverse life events were missing were excluded, which removed 1 (SHP) and 276 (HILDA) person-years. Gender was recorded based on survey-provided categories. In SHP, respondents could select male, female, or other, whereas in HILDA, the options were male and female. In the SHP sample, 47.06% were coded as male, 52.94% as female, and 0.004%

as other; in the HILDA sample, 51.88% as male and 48.13% as female. Computed across person-years, the full sample in SHP had a mean age of 50.04 years (SD = 17.56; median = 50); in HILDA, the mean age was 46.81 years (SD = 17.99; median = 46). The age and gender distribution for the full sample, as well as subsamples observed across 10, 15, and 20 consecutive years, are shown in Supplementary Fig. S1 (SHP) and S2 (HILDA), including the number of education years and working status (SHP) and socio-economic status (HILDA). In SHP, the number of education years is constructed based on the ISCED classification, and in HILDA, socio-economic status is based on the SEIFA 2021 Decile of the Index of relative socioeconomic advantage/disadvantage. The number of observed years per person is shown in Supplementary Fig. S3.

Due to the study design of SHP and HILDA, events were restricted to occurring at most once per year in our analysis. This omitted the within-year event repetition. Though both studies collect data about the event timing, a portion of observations lacked this information (SHP: 18.29%; HILDA: 8.90%). Additionally, HILDA's event timing (recorded by quarter) can be ambiguous when multiple quarters are indicated, making it unclear if events are separate or continuous. This complicates the comparison to SHP, which only records event timing using a single month, such that event repeats are not included.

Whereas household identification across waves is already included in the SHP dataset, HILDA only includes household identification per wave. We created a cross-wave household identification based on whether the same people constituted the same household across waves. No sampling weights were applied. Sampling weights adjust observations to more accurately represent the target population, but this would have complicated the interpretation of the event-level associations we examine.

**Co-occurrence**

To investigate whether one event increases the probability of another event, we used generalized linear mixed-effects models with a binomial distribution and logit link function. For each event, we ran a separate model with each of the

other events serving as predictors, as well as a random intercept for both the individual and the household to account for differences in base event rates. For example, the formula for the first event in the HILDA dataset was "$E_1 \sim E_2 + E_3 + E_4 + E_5 + E_6 + E_7 + E_8 + E_9 + E_{10} + E_{11} + E_{12} + E_{13} + (1|\text{individual}) + (1|\text{household})$", $E$ being event. We conducted a sensitivity analysis examining pairwise associations between events without adjustment for other events (still including the same random effects), to assess whether controlling for all other events influenced the observed associations. The models were fit to 11,3605 (SHP) and 210,031 (HILDA) total observations, consisting of 16,946 (SHP) and 25,803 (HILDA) individuals and 11,231 (SHP) and 29,410 (HILDA) households. Note that the number of households can be larger than the number of individuals, as households were defined as a unique combination of individuals living together, such that an individual leaving or entering counted as a new household.

Due to the design of SHP, personal illnesses and accidents cannot co-occur within the same year, as only one was recorded per year – they are mutually exclusive subcategories. This means that odds ratios will always be zero for combinations of these events. As such, other personal illnesses and accidents were removed as predictors in models with as outcome another personal illness or accident. For an outcome event that is a personal illness or accident, all other personal illnesses or accidents will always be zero, resulting in perfect collinearity of predictors. For this same reason, the "other" category of personal illnesses or accidents was used as a reference category in all models and thus dropped as a predictor.

In addition to the contemporaneous associations, we investigated lagged (1 year) associations between events. The analysis was implemented in the same way as for the contemporaneous associations, except that the outcome event was lagged by one year, and the outcome event type was also entered as a predictor. The models were fit to a median sample of 92,692 (SHP) and 174,957 (HILDA) total observations, consisting of 13,895 (SHP) and 21,738 (HILDA) individuals and 9469 (SHP) and 25,480 (HILDA) households.

Both the contemporaneous and lag-1 co-occurrence analysis adjust for all other events, which has the potential to distort associations, such as when controlling for mediators or colliders[81]. For instance, a major worsening in finances can be an outcome of both being fired and separation from spouse, such that controlling for a major worsening in finances (a collider) can misrepresent the relation between being fired and separation from spouse. To assess the impact of this modelling choice, we compared our fully adjusted models (controlling for all other events) to bivariate models that included only one predictor event at a time, while retaining the same random effects structure for individuals and households and fitting to the same data. To compare associations, we calculated the percentage difference between unadjusted and adjusted odds ratios as: $100 \times (\text{OR}_{\text{unadjusted}} - \text{OR}_{\text{adjusted}})/\text{OR}_{\text{adjusted}}$. In addition, we identified which significant adjusted associations reversed direction when not adjusting for other events (i.e., $\text{OR}_{\text{unadjusted}} > 1$ and $\text{OR}_{\text{adjusted}} < 1$, or vice versa).

## Autocorrelation in event counts
To investigate the lagged relationship between event counts, we used a generalized linear mixed-effects model with a Poisson distribution and log link function. A random intercept for both individual and household was added to account for heterogeneity in base event counts. The individual random slope was dropped because of convergence issues in SHP. As fixed effects, we included the scaled predictors age and age squared to account for the potentially nonlinear relationship between age and event counts[82], as well as the unscaled lagged event counts to investigate the predictive effect of last year's event count. This yielded the formula "events ~ 1 + age + age$^2$ + lag-1 events + (1|individual) + (1|household)". The models were fit to 143,913 (SHP) and 243,312 (HILDA) total observations, consisting of 21,592 (SHP) and 24,973 (HILDA) individuals and 13,480 (SHP) and 33,014 (HILDA) households.

## Accumulation
To investigate the accumulation of event counts, we chose to restrict our sample to individuals who were continuously observed across twenty years,

without missing years in between. Including missing years would either necessitate imputation or counting missing years as having zero events, obscuring the empirical accumulation process. Choosing a longer consecutive time period would exclude more individuals. Some person-years had missing events: For SHP, 7344, 11, 4, and 1 person-years missed 1, 2, 7, and 8 events, respectively. For HILDA, 946, 63, 20, 0, 1, 1, 13, 2, 4, 6, 20, and 12 person-years missed 1 to 12 events, respectively. To maintain a larger sample, missing events were counted as not having occurred. This resulted in a sample of 1,370 (SHP) and 3,700 (HILDA) individuals, with 1,134 (SHP) and 8,628 (HILDA) households and 27,400 (SHP) and 74,000 (HILDA) total observations. To this restricted dataset, we fitted three models: a "bad luck" model (Poisson), in which risk of adverse life events is homogeneous across the population, a frailty model, in which risk is heterogeneously distributed across individuals and households, and a Polya urn model, in which baseline risk is also heterogeneously distributed across individuals and households, but self-reinforcing over time.

We here specify the "bad luck" model as a Poisson model, where the expected number of adverse life events per year for each individual is governed by the same parameter, $\lambda$, representing the population average rate of events. In this model, the variance of the event counts equals the mean, implying there is no overdispersion. Events occur independently and at a constant rate across all individuals and time points, with no clustering or dependence beyond what would be expected by chance.

Frailty models are used in survival analysis to account for unobserved heterogeneity in the risk of an event (i.e., the hazard) and are popular in econometrics, demographics, and biostatistics[83]. In order to model inter-individual variability in risk that cannot be explained by observed covariates, the hazard is multiplied by an individual random effect (i.e., the frailty)[83]. As such, each individual and each household has their own risk parameter $\lambda_i$ drawn from a distribution such as a log-normal. Each year, the number of events is drawn from a Poisson, meaning that events are independent of each other, given the combination of individual and household risk.

Conversely, a Polya urn is a classic example of a self-reinforcing stochastic process[84]. In its simplest form, a Polya urn contains $w$ white balls and $b$ blue balls. At each timestep, a single ball is drawn from the urn, and it is returned to the urn with $c$ additional balls of the same colour. If we take the white balls to represent no event, and the blue balls to represent one event, the initial risk of an adverse life event is $\frac{b}{w+b}$. Clearly, the initial draws greatly influence which colour will come to dominate the urn due to its self-reinforcing nature. To model how this plays out in a population, we here conceptualize each person as having their own urn, from which several balls (events) are drawn per year. To estimate this process, we note that the population distribution generated by a Polya urn converges to a beta-binomial distribution. Formally, this means that for each individual-household combination $i$, the probability $p_i$ of drawing an adverse life event is drawn from a beta distribution with shape parameters $\alpha = b/c$ and $\beta = w/c$. The beta distribution is defined on the interval [0, 1], and defines the probability of adverse life events in the population. When $\alpha = \beta$, the distribution is symmetric and uniform. When $\alpha > \beta$, the distribution is left-skewed, and when $\alpha < \beta$, the distribution is right-skewed. This means that the higher $\alpha$ is relative to $\beta$, the higher the average probability of adverse life events in the population. Given this probability $p_i$, the number $Y_{i,t}$ of adverse life events experienced within one year $t$ follows a binomial distribution with $k$ trials, where $k$ represents the total number of possible events (SHP: $k = 6$, HILDA: $k = 13$). The resulting beta-binomial distribution exhibits overdispersion relative to a simple binomial, capturing the self-reinforcing nature of adverse life event accumulation: individuals who experience more events early have an increased probability of experiencing additional events over time.

A Poisson, frailty, and Polya urn model thus offers three distinct causal mechanisms underlying the accumulation of adverse life events. In a Poisson model, events are independent over time and across the population, as risk is static and homogeneous. In a frailty model, risk is differentially distributed across the population, and is *independent* of the events which already occurred conditioned on the combination of individual and

household risk. In a Polya urn model with random effects, in addition to a heterogeneous distribution of risk across the population, the events *themselves* increase the probability of future adverse life events. In other words, the Poisson and frailty model assume static risk, whereas risk in the Polya urn model is history-dependent and dynamic. Importantly, the self-reinforcement process in the Polya urn model is not deterministic, because all models involve a degree of randomness as they sample events from a distribution given a certain probability.

The Poisson model was specified as a generalized linear model with a Poisson distribution and log link function (i.e., log-normal random effects). This simple model consisted of only a fixed intercept without any random effects, events ~ 1. To extend this model with random effects, the frailty model incorporated a random intercept for both the individual and the household to account for heterogeneity in base event counts. The frailty model was implemented as a generalized linear mixed-effects model with a Poisson distribution and log link function (i.e., log-normal distributed random effects), with the formula "events ~ 1 + (1|individual) + (1|household)". Finally, to model self-reinforcement, the Polya urn model was implemented as a generalized linear mixed-effects model with a beta-binomial distribution and logit link function. glmmTMB follows Morris' (1997) parameterization of the beta-binomial, in which the variance of event counts is $V = \mu(1 - \mu)(n(\phi + n)/(\phi + 1))$, with $\mu = \frac{\alpha}{\alpha + \beta}$, $\phi = \alpha + \beta$, and $n$ being the number of trials[85]. As the beta-binomial requires both the number of successes and the number of failures, successes were defined as the number of events that occurred, and failures were defined as the number of possible events (SHP: 6; HILDA: 13) minus successes. The model consisted of a fixed intercept with a random intercept for both individual and household to account for heterogeneity in base event counts, yielding the formula "(events, no events) ~ 1 + (1|individual) + (1|household)". For robustness, we repeated the same analysis for 10 and 15 consecutively observed years. For 10 years, this resulted in a sample of 3535 (SHP) and 8,137 (HILDA) individuals, with 2828 (SHP) and 13,216 (HILDA) households and 35,350 (SHP) and 81,370 (HILDA) total observations. For 15 years, the sample consisted of 2620 (SHP) and 4980 (HILDA) individuals, with 2146 (SHP) and 9881 (HILDA) households and 39,300 (SHP) and 74,700 (HILDA) total observations.

Finally, to describe the cumulative distribution of adverse life events, we used the poweRlaw package (v1.0.0) to fit a Poisson, exponential, log-normal, and power-law distribution[54,86,87]. A heavy-tailed distribution is only fitted to a range of the data starting at $x_{min}$, which is estimated by minimizing the Kolmogorov-Smirnov statistic. Each distribution is fitted using maximum likelihood estimation. We compared the fit of the log-normal model to the three alternative models using Vuong's likelihood ratio test[88]. A positive test statistic indicates the log-normal model fits better, whereas a negative value favours the alternative model. The null hypothesis is that both distributions are equally far from the true distribution, which we assessed using a two-sided *p*-value. To compare two models, both need to be estimated using the same $x_{min}$, for which we chose the median $x_{min}$ of the two models. To fit these distributions, people with zero cumulative adverse life events had to be excluded, which removed 1 (SHP) and 19 (HILDA) people.

**Implementation**
All analyses were implemented in R[89], (v4.4.1). Full analysis scripts are available at GitHub (https://github.com/KCEvers/PatternsAdverseLife Events)[90]. All generalized linear mixed-effects models were fit using the *glmmTMB* function from the glmmTMB package (v1.1.13) with estimation via maximum likelihood and default nonlinear optimizer nlminb[91]. To assess model fit, deviations from model assumptions were checked using the performance (v0.15.2) and DHARMa package (v0.4.7)[92,93]. Specifically, models were assessed for multicollinearity, non-normality of residuals, and non-normality of random effects[94,95]. All statistical tests were two-sided, and 95% profile confidence intervals were used to assess significance, computed using the *confint* function from base R. The marginal and conditional explained variance ($R^2$) and the intra-class correlation coefficient (ICC)[96] for each model were computed with the performance package (v0.15.2)[93]. These

compute the variance explained by fixed effects only, the variance explained by both fixed effects and random effects, and the proportion of variance attributable to clustering (individual/household grouping) after accounting for fixed effects. In models without fixed effects except for the intercept, the marginal $R^2$ is zero, and the adjusted ICC is equal to the conditional $R^2$. In the accumulation analysis, models were compared using the Akaike Information Criterion (AIC) and the Bayesian Information Criterion (BIC). Both metrics are appropriate for non-nested model comparisons[97,98]. AIC emphasizes predictive accuracy while BIC imposes stronger penalties for model complexity. For clarity, we report fit comparisons as ΔAIC = AIC(focal model) − AIC(comparison model) and similarly for BIC; a positive difference indicates the comparison model fits better.

**Ethics information**
This study involved a secondary analysis of de-identified data from the SHP and HILDA surveys. Both studies obtained informed consent from participants and received ethics approval from their respective institutional review boards. The SHP data is available free of charge for the entire research community. SHP is integrated into the Swiss Centre of Expertise in the Social Sciences (FORS) and hosted by the University of Lausanne. Access to HILDA was approved by Longitudinal Studies, Australian Government Department of Social Services. This study complied with the Terms of Use of SHP and DSS Longitudinal Studies Data Access and Use Guidelines of HILDA. As the SHP and HILDA data are publicly available, and confidentiality is protected by identity masking, this study is exempt from ethical assessment by the institutional review board of the University of Amsterdam. No local researchers were included as this study concerned a secondary analysis of publicly available data. The analyses outlined in this study were not pre-registered.

## Results
### Event types
**Base rates of event types**. The relative frequency of adverse life events across all person-years is shown in Fig. 1. Dashed vertical lines indicate 10000 instances, where for example, events surpassing the third dashed line occurred for over 30,000 person-years. Overall, rates of common adverse life events are similar across SHP and HILDA datasets, but show different patterns in event frequencies in part due to differences in measurement and methodology. The most common events in both datasets are the injury, illness, or accident of a closely related person, followed by the death of a closely related person, which together were responsible for 54.01% (SHP) and 40.58% (HILDA) of all events. In SHP, the next most common events were more similar in frequency (9.76 - 5.25%): physical illness, conflicts with or among related persons, problems with own children, termination of close relationship, followed by unspecified events. SHP thus focused more on the interpersonal domain, whereas HILDA also included events involving housing, financial, and legal matters. The least common adverse life events in HILDA were a natural disaster, the death of a spouse or child, and being detained in jail.

**Co-occurrence**. Adjusted odds ratios (OR) of all adverse life events within the same year are shown in Fig. 2 (HILDA) and Fig. S4 (SHP). OR with confidence intervals that excluded one are coloured, ranging from lower (yellow) to higher (red) OR. An OR above 1 indicates that the outcome event is more likely if the predictor event occurred compared to if it did not. For example, an OR = 2 indicates that the outcome event is twice as likely to occur if the predictor event also occurred within that year. Here we present adjusted OR (with 95% confidence intervals in square brackets), which account for the effect of other events.

Both datasets show that associations between events are remarkably common. Most associations were between 1 < OR < 2, but a sizeable group of event combinations showed substantial associations, many of which were intuitive. In HILDA, the strongest associations were between being detained in jail (A) and having a close family member detained in jail (B) ($OR_{A \to B} = 16.77$ [9.87, 28.49], $OR_{B \to A} = 12.02$ [8.04, 17.96]) and being fired

**Fig. 2 | Adjusted odds ratios (OR) describing the contemporaneous associations between all adverse life events (HILDA).** The estimates are based on 25,803 individuals from 29,410 households with a total of 210,031 person-years (Source: Household, Income and Labour Dynamics in Australia, HILDA). OR are adjusted for the effects of other variables, and are coloured from low (yellow) to high (red), which is only shown if the 95% profile confidence interval excludes an OR of one. An OR above one indicates the predictor event increased the odds of the outcome, which was the case for all significant events.

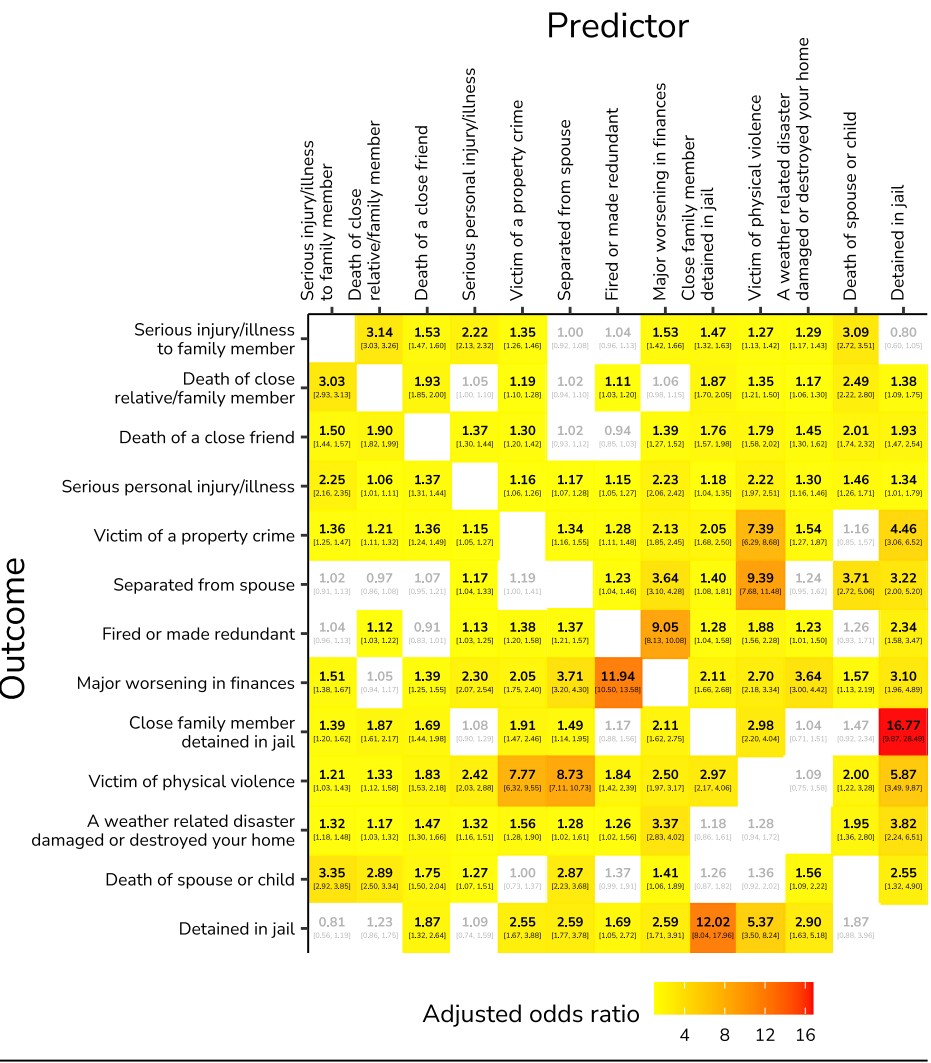

or made redundant (A) and a major worsening in finances (B) (OR$_{A\rightarrow B}$ = 11.94 [10.50, 13.58], OR$_{B\rightarrow A}$ = 9.05 [8.13, 10.08]). Strong associations were also found between being a victim of physical violence (A) and being a victim of a property crime (B) (OR$_{A\rightarrow B}$ = 7.39 [6.29, 8.68], OR$_{B\rightarrow A}$ = 7.77 [6.32, 9.55]) and between being a victim of physical violence (A) and separation from spouse (B) (OR$_{A\rightarrow B}$ = 9.39 [7.68, 11.48], OR$_{B\rightarrow A}$ = 8.73 [7.11, 10.73]).

In SHP (Fig. S4), the strongest association was between the termination of a close relationship (A) and conflicts with or among related persons (B) (OR$_{A\rightarrow B}$ = 4.94 [4.57, 5.34], OR$_{B\rightarrow A}$ = 5.00 [4.64, 5.39]). Other associations were smaller, such as between problems with own children (A) and conflicts with or among related persons (B) (OR$_{A\rightarrow B}$ = 2.53 [2.38, 2.69], OR$_{B\rightarrow A}$ = 2.52 [2.37, 2.67]), between problems with own children (A) and the illness or accident of a closely related person (B) (OR$_{A\rightarrow B}$ = 2.34 [2.24, 2.44], OR$_{B\rightarrow A}$ = 2.33 [2.23, 2.43]), between mental illness (A) and conflicts with or among related persons (B) (OR$_{A\rightarrow B}$ = 2.09 [1.70, 2.58]), or mental illness (A) and the termination of a close relationship (OR$_{A\rightarrow B}$ = 2.03 [1.60, 2.58]).

The sensitivity analysis comparing adjusted and unadjusted odds ratios showed that all significant associations maintained the same direction, with unadjusted estimates being larger than adjusted estimates (SHP: mean = 4.61%, median = 2.80%, SD = 6.58; HILDA: mean = 43.16%, median = 29.38%, SD = 38.69; Supplementary Figs. S5, S6). This pattern suggests that controlling for other events primarily reduced the magnitude of associations, but did not change their direction.

Most event associations are quite intuitive and may even refer to the same event. For example, the association between being fired or made

redundant and a major worsening in finances could trivially refer to the immediate loss in finances after losing one's job. This same concern holds for event combinations like a serious injury/illness to a family member and the death a family member, being a victim of physical violence and a victim of a property crime, and conflicts with or among related persons and problems with own children. As events are aggregated per year, it is not possible to tell how much time has passed between them. As such, we cannot tell to what degree they are related and refer to the same event, or whether they are temporally separated enough to count as separate events. This is a common methodological concern with event checklists (see *Discussion* section). However, joint and conditional probabilities show that event co-occurrence is quite low (Supplementary Figs. S11, S13, S15, and S17), the highest joint probability being 0.09 (SHP) and 0.04 (HILDA). These low joint and conditional probabilities show that event associations do not merely arise from referring to the same event, as the mean conditional probability of, for example, being fired or made redundant and a major worsening in finances is only 0.17.

Overall, conditional probabilities are quite asymmetrical due to the different base rates of events. For instance, being jailed showed a strong asymmetrical conditional probability with all other adverse life events, where being jailed increases the probability of other events, but other events do not substantially increase the probability of being jailed (Supplementary Fig. S17).

In all models, fixed effects accounted for minimal explained variance (i.e., low marginal $R^2$), whereas adding random effects increased explained variance substantially (i.e., much higher conditional $R^2$; see Supplementary Tables S1 and S2). This means that other events contributed only

**Fig. 3 | Adjusted odds ratios (OR) describing the lag-1 associations between all adverse life events (HILDA).** The estimates are based on a median sample of 21,738 individuals from 25,480 households with a total of 174,957 person-years (Source: Household, Income and Labour Dynamics in Australia, HILDA). OR are adjusted for the effects of other variables, and are coloured from low (yellow) to high (red), which is only shown if the 95% profile confidence interval excludes an OR of one. An OR above one indicates the predictor event increased the odds of the outcome event the next year, and an OR below one indicates the odds were reduced.

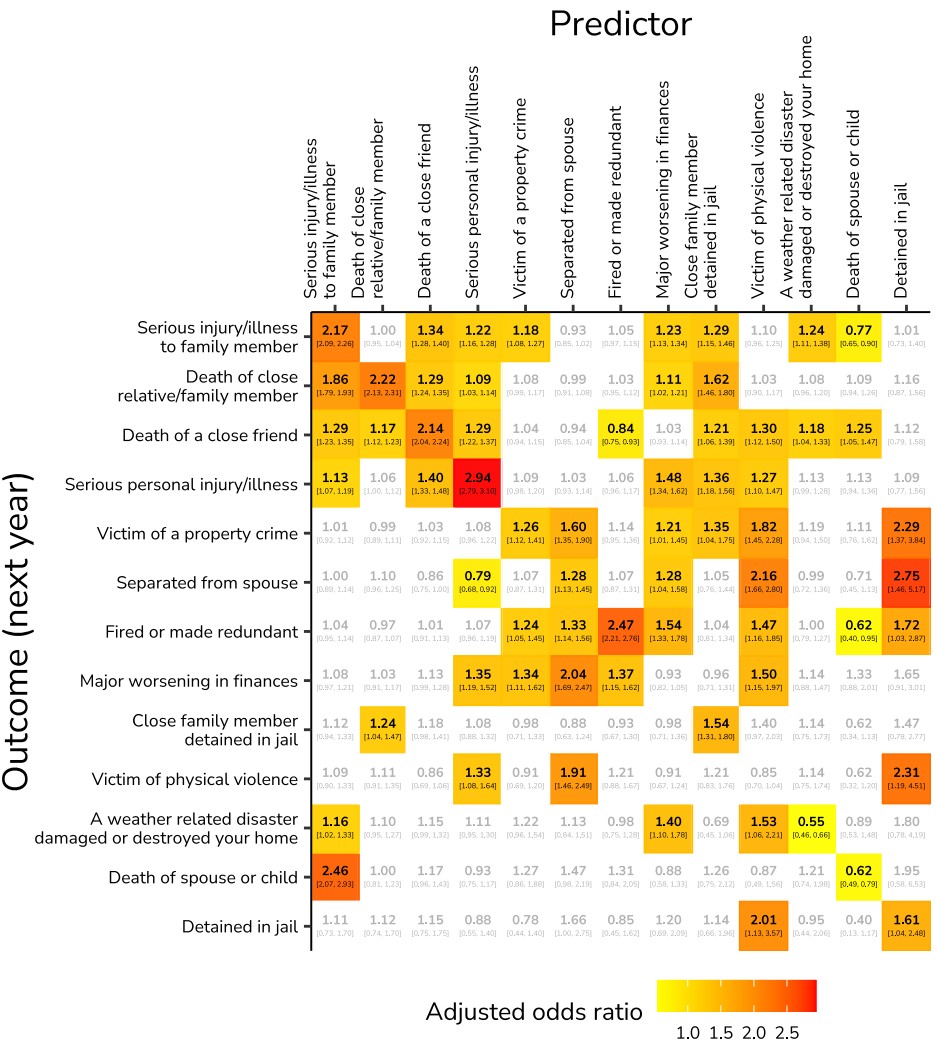

marginally, whereas unobserved heterogeneity between individuals and households accounted for a much larger part of the explained variance. This was particularly the case for rare events such as being jailed or a work accident, where random effect distributions were distinctly non-normal and in many cases bi-modal. As such, the variance of the random intercepts for these events was profoundly inflated and not reliable.

**Lag-1 co-occurrence.** Adjusted odds ratios (OR) for lagged (1 year) relationships are presented in Fig. 3 (HILDA) and Supplementary Fig. S7 (SHP). These show the odds of an outcome event given that a predictor event happened the previous year. Similar to the contemporaneous associations, lag-1 associations were widespread, and the majority was between 1 < OR < 2. In HILDA, among the strongest associations were relationships within the event type itself. For example, a serious personal injury or illness increased the odds of experiencing a serious personal injury or illness again the next year by OR = 2.94 [2.79, 3.10]. Similarly strong within-event type relationships were found for being fired or made redundant (OR = 2.47 [2.21, 2.76]), for separation from spouse (OR = 3.05), death of a close relative or family member (OR = 2.22 [2.13, 2.31]), a serious injury or illness to a family member (OR = 2.17 [2.09, 2.26]), and the death of a close friend (OR = 2.14 [2.04, 2.24]). Notable associations between events included a serious injury or illness to a family member increasing the odds of the death of a spouse or child the year after (OR = 2.46 [2.07, 2.93]), separating from spouse increasing the odds of a major worsening in finances the year after (OR = 2.04 [1.69, 2.47]), and being a victim of physical violence increasing the odds

of separating from spouse the year after (OR = 2.16 [1.66, 2.80]). Moreover, being detained in jail increasing the odds of separating from spouse (OR = 2.75 [1.46, 5.17]), being a victim of physical violence (OR = 2.31 [1.19, 4.51]), and being a victim of a property crime (OR = 2.29 [1.37, 3.84]) the year after.

Similarly, in SHP, among the strongest associations were relationships within the same event type. The strongest within-event type relationships included physical illness (OR = 3.21 [3.03, 3.41]), conflicts with or among related persons (OR = 2.69 [2.50, 2.89]), the termination of a close relationship (OR = 2.60 [2.35, 2.87]), and problems with own children (OR = 2.32 [2.19, 2.45]). Between events, a road accident increased the odds of a work accident the next year (OR = 4.25 [1.34, 13.50]), a sport accident increased the odds of a road accident the next year (OR = 2.70 [1.45, 5.04]), a work accident increased the odds of an accident in the home or garden the next year (OR = 2.46 [1.03, 5.84]), and conflicts with or among related persons increased the odds of the termination of a close relationship the next year (OR = 2.34 [2.15, 5.56]). All other associations were below an OR of two.

Unlike the contemporaneous associations, a number of lag-1 associations showed *reduced* odds. In HILDA, the death of a spouse or child decreased the odds of the death of a spouse or child (OR = 0.62 [0.49, 0.79]), being fired or made redundant (OR = 0.62 [0.40, 0.95]), or a serious injury or illness of a close family member (OR = 0.77 [0.65, 0.90]) the next year. In addition, a weather-related disaster damaging or destroying one's home decreased the odds of the same event happening the next year (OR = 0.55 [0.46, 0.66]). A serious personal injury or illness reduced the odds of

**Fig. 4 | Accumulation of adverse life events across twenty consecutive years.** Accumulation analysis based on a sample of 1370 individuals and 1134 households (Source: Swiss Household Panel, SHP) and 3700 individuals and 8628 households (Source: Household, Income and Labour Dynamics in Australia, HILDA). The empirical data is shown in red dots, with the fit of the Poisson, frailty, and Polya urn models shown in blue, green, and purple, respectively (1,000 model simulations). Notably, the Poisson ("bad luck") model shows a substantial misfit to the empirical data, both underestimating the frequency of a low and high number of events. In contrast, the frailty and Polya urn model fit the empirical data very well, and are therefore mostly obscured by the empirical data. To increase visibility, the x-axis is restricted to the range of the empirical data, which does not show that the frailty model predicts a much heavier tail than empirically observed (see Supplementary Fig. S19).

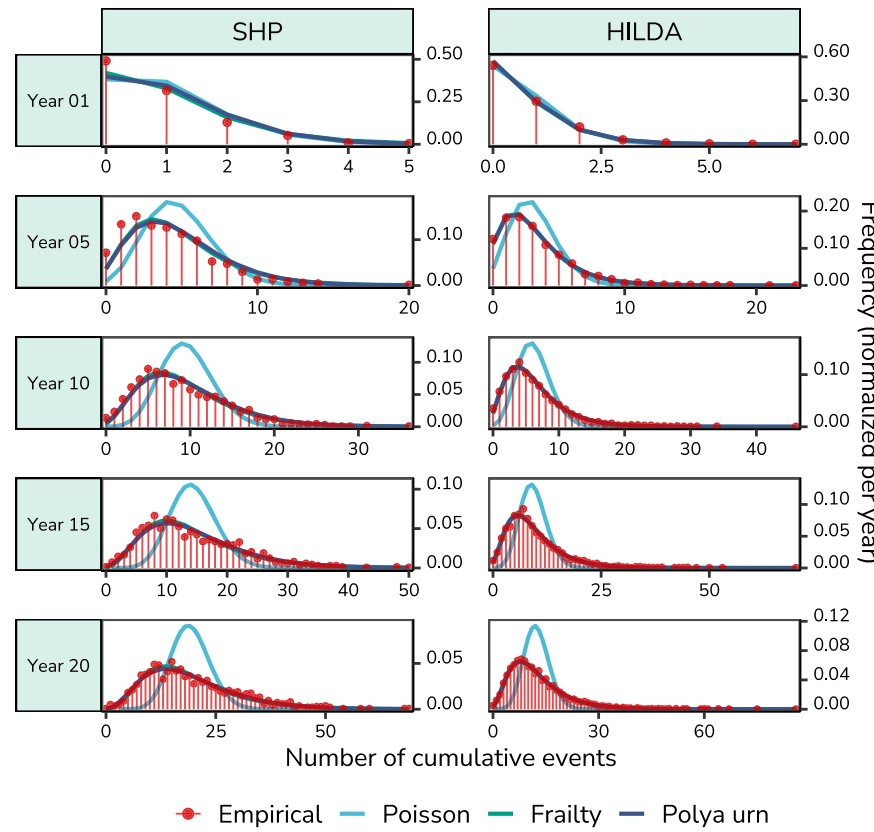

separating from a spouse the year after (OR = 0.79 [0.68, 0.92]), and being fired or made redundant decreased the odds of the death of a close friend the next year (OR = 0.84 [0.75, 0.93]). In SHP, the only significant reduced association was between a sports accident reducing the odds of an unspecified illness or accident the next year (OR = 0.68 [0.50, 0.93]).

The sensitivity analysis for lag-1 associations showed similar patterns as the contemporaneous analysis, with unadjusted estimates being larger than adjusted estimates (SHP: mean = 1.27%, median = 1.24%, SD = 11.37; HILDA: mean = 10.40%, median = 7.08%, SD = 14.43; Supplementary Figs. S8-S9). All significant adjusted associations maintained the same direction when unadjusted for other events, except one: the death of a spouse or child in HILDA, which changed from a negative adjusted association (OR = 0.62 [0.49, 0.79]) to a positive unadjusted association (OR = 1.55 [1.27, 1.89]).

Compared to the contemporaneous associations, lag-1 associations were weaker, less common, and more asymmetrical. The lag-1 joint and conditional probabilities were also generally moderately weaker than their contemporaneous counterparts (Supplementary Figs. S12, S14, S16, and S18).

**Summary**. To briefly summarize, we first observed that over half of the time, no adverse life event happens in a year. If an event occurs, it co-occurs with another event in about 40% of observations, with a decreasing probability of a higher number of co-occurring events. When investigating the relations between events within the same year, positive associations appear to be the norm rather than the exception. That is, all significant odds ratios were above one, indicating that the occurrence of one event *increased* the probability of another event occurring. Most associations are small in magnitude (1 < OR < 2), but a select number of event associations are considerably larger. Associations were found between and across independent and dependent events. Similarly, lag-1 associations were also common, though generally less so, weaker, and more asymmetrical than contemporaneous associations, including some negative associations.

## Event counts

**Yearly event count distribution**. Further structural patterns in adverse life events may be found in event counts, where we simply count the number of adverse life events per year, collapsing across event types. As shown in the top panels of Fig. 4, for over half of the observations (SHP: 41.07%; HILDA: 56.26%), no adverse life events occur in a year. One event occurs for 33.49% (SHP) and 28.24% (HILDA) of observations, and events co-occur (2 or more events) for only 25.45% (SHP) and 15.50% (HILDA) of observations. Viewed another way, if an event occurs, in 43.18% (SHP) of 35.43% (HILDA) of observations, it co-occurs with another event. This distribution of co-occurrences was largely consistent throughout the years (Fig. S10).

**Autocorrelation in event counts**. Lagged event counts were positively associated with event counts the next year (Table 1), such that each additional event increased the rate of events next year by 17.71% (SHP) and 16.69% (HILDA). Similarly, age was positively associated with event counts, meaning that older people experienced more events. The squared effect of age was small and non-significant in SHP, and small and significant in HILDA, indicating the weak presence of a non-linear relationship between age and event count in HILDA. The base rate of event counts (i.e. the intercept) varied substantially per individual and household. In fact, random effects accounted for a considerably larger proportion of variance in adverse life event counts than fixed effects (SHP: Marginal $R^2$ = 0.043, Conditional $R^2$ = 0.203; HILDA: Marginal $R^2$ = 0.027, Conditional $R^2$ = 0.274). They also accounted for a moderate amount of the overall variance after adjusting for fixed effects (adjusted ICC, SHP: 0.167, HILDA: 0.254).

**Accumulation**. To investigate what the autocorrelation in adverse life event counts amounts to across decades, we present the cumulative distribution of the number of adverse life events across twenty years in red in Fig. 4. The distribution at twenty years has a range of 0 to 73 (SHP) and 1 to 86 (HILDA) events, a mean of 19.70 (SHP) and 12.46 (HILDA)

**Table 1 | Model estimates of the autocorrelation in event counts**

| | Fixed | | | | Random $\sigma$ | |
|---|---|---|---|---|---|---|
| | $\lambda$ | Lag-1 counts | Age | Age$^2$ | $i$ | $h$ |
| SHP | 0.72 [0.71, 0.73] | 1.18 [1.17, 1.18] | 1.10 [1.09, 1.11] | 1.00 [1.00, 1.01] | 0.27 [0.25, 0.28] | 0.27 [0.26, 0.28] |
| HILDA | 0.48 [0.47, 0.48] | 1.17 [1.17, 1.18] | 1.11 [1.10, 1.12] | 1.02 [1.01, 1.02] | 0.40 [0.38, 0.41] | 0.39 [0.38, 0.40] |

Estimates are on the natural scale (95% profile confidence intervals in brackets). $\lambda$ = Yearly rate of adverse life events; $i$ = Individual; $h$ = Household (Source: Swiss Household Panel, SHP and Household, Income and Labour Dynamics in Australia Survey, HILDA).

events, a median of 17 (SHP) and 11 (HILDA) events, and a coefficient of variation of 5.77 (SHP) and 5.57 (HILDA), indicating a high level of dispersion relative to the mean. This is reflected by the distributions' remarkably heavy tails, meaning some people experience a very large number of adverse life events, whereas the majority of people experience far fewer events. The heavy-tailed distribution is best described by a log-normal or exponential distribution (see Supplementary Fig. S20 and Supplementary Table S7). Across the years, we thus observe a growing disparity in the number of adverse life events that people experience, which corresponds to the autocorrelation of adverse life event counts.

The empirical distributions are remarkably different from those that would arise from a "bad luck" process (Poisson), in which the risk of adverse life events is the same for all individuals, meaning that events are randomly distributed across the population and across time (shown in blue in Fig. 4). Although the Poisson distribution overlaps reasonably well with the empirical distribution in the first year, after twenty years, the Poisson model considerably underestimates the frequency of a very low number of events, as well as the frequency of a very high number of events. It predicts a symmetrical distribution without heavy tails and quite a narrow range of events. The poor fit of the Poisson distribution follows straight-forwardly from the observed widespread contemporaneous and lag-1 associations between events. Though deviations from the Poisson are expected in case events were double-counted, low conditional probabilities show this is only a minor concern (see *Co-occurrence* section). The non-randomness of adverse life events thus becomes particularly apparent by how they accumulate across decades, demonstrating that *who* gets events is not simply a matter of chance or "bad luck".

As a random distribution process of events fits the data poorly, we considered two alternative stochastic processes which do include dependencies: a frailty model, in which risk is heterogeneously distributed across the population but random across time, and a Polya urn model, in which baseline risk is also heterogeneously distributed across the population, but the risk of events is self-reinforcing over time (shown in green and purple in Fig. 4). Model estimates and fit comparisons are shown in Table 2. The frailty and Polya urn model both indicated a higher population risk of adverse life events for SHP than HILDA, with a higher rate of events (SHP: $\lambda = 0.86$; HILDA: $\lambda = 0.52$) and a greater proportion of events compared to no events in the urn (SHP: $\alpha = 25.57$, $\beta = 150.47$; HILDA: $\alpha = 37.13$, $\beta = 884.75$). Both models fit exceedingly better than the Poisson model, but the Polya urn model fit the best (Table 2). These findings were robust to limiting the consecutive observation period to 10 and 15 years (Supplementary Tables S5 and S6). Though the frailty and Polya urn model generate very similar distributions, the frailty model predicts a much longer tail than shown by the empirical data (Supplementary Fig. S19). As such, the accumulation process of adverse life events seems to be better described by a self-reinforcing process than a heterogeneous risk process.

Similarly, the magnitude of the explained variance accounted for by random effects was considerably larger in the Polya urn than the frailty model, shown by the larger adjusted ICC of the Polya urn model (SHP: 0.92; HILDA: 0.97) compared to that of the frailty model (SHP: 0.29; HILDA: 0.28). This indicates that individual and household heterogeneity accounted for much more of the explained variance in the Polya urn model than in the frailty model. The improved fit of the Polya urn model compared to the frailty model is thus probably due to the restriction in the number $k$ of possible events in the Polya urn model, as well as the high dispersion in the

risk distribution, as the Polya urn can account for this large heterogeneity using a separate dispersion parameter.

**Summary.** To summarize the findings on adverse life event counts, lagged event counts were positively associated, such that each additional event increased the rate of events next year by around 17%. Across twenty years, the cumulative distribution of event counts displays overdispersion and heavy tails, with a growing difference in event count between those who experience few events and those who experience many. The cumulative distribution is distinctly different from a "bad luck" model, in which the risk of adverse life events is homogeneous across the population. A frailty model in which risk is heterogeneously distributed across individuals and households fits much better, but the best-fitting model in both datasets is a Polya urn model, which shows self-reinforcement of risk over time.

## Discussion
In summary, the present study demonstrated in two panel datasets that adverse life events show a substantial degree of non-randomness. Our findings on the co-occurrence and accumulation of adverse life events offer evidence of their non-random nature from different angles. We identified widespread contemporaneous and lagged associations between adverse life events. Though contemporaneous relationships between events at a yearly level are not informative of their temporal order or degree of connectedness, they clearly show that events do not co-occur as a matter of "bad luck". Lagged associations between event types as well as event counts further underscore the relations between events over time, wherein adversity begets adversity. The accumulation analysis indicates that the risk of adverse life events is best modelled as heterogeneously distributed across the population with an additional self-reinforcement process, such that the risk of adverse life events depends on one's history of adversity.

## Limitations
For all analyses, differences between individuals and households were stronger predictors of event occurrence than concurrent or prior adverse life events. That is, individual- and household-level characteristics as captured by random effects accounted for a much larger part of the explained variance than fixed effects. This aligns with genetically informed research demonstrating substantial genetic influence on adverse life event exposure and reporting[29], described as "pervasive in extent and modest to moderate in impact"[22], with genetic factors influencing exposure through personality traits, behavioural tendencies, and other stable characteristics. Such variables are captured by the random intercepts in our models, which model unobserved variables that elevate overall adverse life event risk. However, the random intercepts cannot account for unmeasured characteristics that may differentially influence specific types of event sequences beyond general elevation in risk. For instance, while impulsivity may increase baseline event rates generally (captured by the random intercept), it might also specifically increase both financial problems and subsequent relationship conflicts more than other event pairs, potentially creating spurious lag-1 associations between these particular events. Genetically informed designs, such as co-twin control studies comparing associations between versus within twin pairs, could help disentangle trait-mediated associations from genuine event-to-event effects[29]. Another important confounder to consider is events themselves, as individuals experiencing the most events may be more

**Table 2 | Estimates of the Poisson, frailty, and Polya urn models in the accumulation analysis across 20 years**

| | Poisson | Frailty | | | Fit (Poisson-Frailty) | | Polya urn | | | | Fit (Frailty-Polya) | |
| | Fixed | Fixed | Random σ | | | | Fixed | | Random σ | | | |
| | $\lambda$ | $\lambda$ | $i$ | $h$ | ΔAIC | ΔBIC | $p$ | $\phi$ | $i$ | $h$ | ΔAIC | ΔBIC |
|---|---|---|---|---|---|---|---|---|---|---|---|---|
| SHP | 0.96 [0.95, 0.97] | 0.86 [0.83, 0.88] | 0.34 [0.31, 0.39] | 0.36 [0.32, 0.40] | 3759.06 | 3742.62 | 0.15 [0.14, 0.15] | 176.03 | 0.43 [0.38, 0.49] | 0.43 [0.38, 0.49] | 494.87 | 486.66 |
| HILDA | 0.62 [0.61, 0.63] | 0.52 [0.51, 0.53] | 0.47 [0.45, 0.50] | 0.37 [0.35, 0.40] | 9483.40 | 9464.98 | 0.04 [0.04, 0.04] | 921.88 | 0.41 [0.39, 0.44] | 0.41 [0.39, 0.44] | 132.57 | 123.36 |

Estimates are on the natural scale (95% profile confidence intervals in brackets). Fit comparisons subtract the fit of the model on the left from that of the model on the right, such that a positive number indicates the model on the left fits better. $\lambda$ Yearly rate of adverse life events, $p$ Per-trial probability of adverse life events, $\phi$ Dispersion, $i$ Individual, $h$ Household, AIC Akaike Information Criterion, BIC Bayesian Information Criterion (Source: Swiss Household Panel, SHP and Household, Income and Labour Dynamics in Australia Survey, HILDA).

likely to drop out of the study. Selective attrition would be expected to underestimate the self-reinforcing effects of adverse life events. Given these limitations, our findings are best interpreted as descriptive patterns, rather than reflecting direct causal effects between events.

The used adverse life event data consists of self-reported event checklists, with several inherent limitations. In contrast to interview-based measures, the respondent largely decides what defines an adverse life event (such as with an illness) and whether a complex event is counted as one or multiple events (or rather as a chronic stressor)[17,99]. Event checklists administered yearly obscure how multiple events are related to each other. It may be intuitive to assume that, for example, two events such as being fired and a worsening of finances or two events like a personal conflict and the end of a relationship directly follow each other, but these may instead be largely separated in time, or refer to different events entirely. Moreover, these decisions may be influenced by the effects of the stressor[100], such that event checklists capture an amalgamation of stressor exposure, response, and outcome[101]. For instance, adverse life events can precipitate mental health problems such as depression and anxiety[66,67,102], which in turn may influence subsequent event reporting through mechanisms such as heightened negative memory bias increased rumination on events[103,104]. This creates potential for spurious lag-1 associations wherein mental health consequences of events in one year inflate the reporting of events in the subsequent year, independent of actual event occurrence. In fact, the concordance between event checklists and interview-based measures appears to be less than 40% (for an overview, see[17]). Event checklists thus potentially contain large intra-categorical variability within the same event type[105,106]. Some event frequencies also deviated considerably from robust estimates, such as the prevalence of mental illness[107], which is in part due to the survey design. Another limitation of our accumulation analysis is the inability to include adversities beyond the scope of the data set, such as childhood adversities, which could therefore not contribute to the self-reinforcing process. For further discussion on stress measurement, we refer the reader to[17,40,99,108].

Keeping these limitations in mind, we briefly discuss the implications of structural patterns in adverse life events in two domains: conceptualization and prevention.

## Conceptualization

Associations between independent and dependent events challenge our conceptualization of their respective causes, which may not be as distinct as generally assumed. Rather than reflecting causal links, these associations could arise from the crude way events are categorized in checklists as either independent or dependent. For example, an event such as a road accident could categorically be classified as independent, obscuring the fact that personal factors such as recklessness could largely have been responsible. However, beyond classification issues, these associations may also reflect more substantive mechanisms, such as common causes or reciprocal causal relations. Common causes complicate the distinction between independent and dependent events, as they show that dependent events may be shaped by external factors beyond an individual's personal characteristics, while some independent events are not as "fateful" after all. For example, becoming ill more often (typically categorized as independent) as well as higher interpersonal stress (typically categorized as dependent) are both related to neuroticism[109,110]. Similarly, reciprocal causal pathways between independent and dependent events demonstrate that their respective causes may not be as distinct. If independent events can increase the probability of dependent events and vice versa, personal characteristics and behaviours are both partially the cause and effect of adverse events. As such, the distinction between independent and dependent events may not be as clear-cut.

A different conceptualization of adverse life events calls for a shift in perspective: from isolated to interconnected events[111], from static to dynamic processes unfolding over time[59], from a separation between person and environment to a transactional approach. In turn, our conceptualization of adverse life events bears on other theories in which such events play a causal role. For example, the distinction between independent and dependent adverse life events is used to differentiate diathesis-stress from stress generation models[112,113]. In diathesis-stress models, a certain disease or

disorder, for example depression, is thought to arise from the interaction between a (genetic) vulnerability and external stressors, where stressors – whether independent or dependent – act as triggers for those with high vulnerability. In stress generation, those vulnerable to depression actively contribute to stressor exposure, particularly dependent events. The differential association between depression and independent versus dependent events is then used as evidence for one model over the other[112]. However, such a test is problematic in the case of causal relations between independent and dependent events, as stress generation models would no longer only be associated with dependent events, but also to independent events due to common causes or reciprocal causal relations. In this case, a transactional model would be a better alternative, which tests the bidirectional influence of independent and dependent stressors over time[112].

## Prevention strategies

Self-reinforcing pathways between events offer opportunities for targeted interventions to prevent adversity from escalating. However, intervening effectively requires identifying self-reinforcing pathways between events and how to disrupt them. For instance, loneliness appears to be the biggest challenge after losing one's spouse[114–116], and forms a straight-forward self-reinforcing pathway to other adverse life events. It is strongly linked to morbidity and mortality[117], shows a reciprocal relationship to depressive symptoms[117,118], and increases health-risk behaviours such as smoking and low physical activity[119]. Similarly, losing one's job can activate other self-reinforcing pathways, where for example moving to a poorer neighbourhood may exacerbate financial stress, social isolation, and mental health difficulties[6]. Recognizing these pathways allows for interventions beyond the individual, targeting environmental factors that sustain adversity. Family-based programs, for instance, have been shown to reduce the negative effects of parental incarceration on children[120], including antisocial behaviour[121], delinquency, marital separation, and mental health disorders[122]. Targeting pathways between events involving both the individual and their environment offers a powerful preventative measure against the spread of adversity.

Most of all, self-reinforcing processes underscore the need for *early* preventative efforts to avert the ensuing adversity cascade. Childhood adversity, which includes all physical and emotional abuse or neglect[123], is highly common, with estimates of around 63.9% of the US experiencing at least one adverse childhood experience, and 17.3% experiencing four or more[124]. Addressing childhood adversity is widely recognized as one of the most effective strategies for preventing morbidity and mortality[125–131]. For example, childhood adversity is estimated to account for 29.8% of all mental disorders worldwide[132], 27.0% of chronic obstructive pulmonary disease[128], 52.0% of violence perpetration[129], and 58.7% of heroin/crack cocaine use[129]. The self-reinforcing effects of childhood adversity are thought to be mediated by maladaptive coping styles[133], engaging in health-harming behaviours, such as smoking, alcohol and drug use[129,134], criminal behaviour[135], negative affect states such as depression, anger, and hostility[133,136], as well as numerous disruptions to nervous, endocrine, and immune systems[137,138]. Crucially, the multitude of activated self-reinforcing pathways pose significant challenges for intervention later in life, underscoring the need for early prevention.

## Data availability

The SHP data is available at https://www.swissubase.ch/en/catalogue/studies/6097/20179/overview[78], with instructions for download at https://forscenter.ch/projects/swiss-household-panel/data. Access to the HILDA dataset can be requested at https://dataverse.ada.edu.au/dataverse/DSSLongitudinalStudies[79].

## Code availability

All analysis scripts are available on GitHub (https://github.com/KCEvers/PatternsAdverseLifeEvents)[90].

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

## Acknowledgements

This study is part of the project 'New Science of Mental Disorders' (https://www.nsmd.eu), supported by the Dutch Research Council and the Dutch Ministry of Education, Culture and Science (NWO gravitation Grant Number 024.004.016). Denny Borsboom is supported by Dutch Research Council Grant VI.C.181.029. This study has been realised using data collected by the Swiss Household Panel (SHP), which is based at the Swiss Centre of Expertise in the Social Sciences FORS. The project is supported by the Swiss National Science Foundation. In addition, this paper uses unit record data from Household, Income and Labour Dynamics in Australia Survey [HILDA] conducted by the Australian Government Department of Social Services (DSS). The findings and views reported in this paper, however, are those of the author[s] and should not be attributed to the Australian Government, DSS, or any of DSS' contractors or partners. DOI: 10.26193/NBTNMV. The funders had no role in study design, data collection and analysis, decision to publish or preparation of the manuscript.

## Author contributions

K.E.: conceptualization (lead), data curation, formal analysis, methodology, software, visualization, writing – original draft preparation, writing – review & editing. D.B.: conceptualization, funding acquisition, supervision, writing – review & editing. E.F.: conceptualization, funding acquisition, supervision, writing – review & editing. F.H.: conceptualization, supervision, writing – review & editing. F.B.: conceptualization, formal analysis, methodology, writing – review & editing. L.W.: conceptualization, funding acquisition, supervision (lead), writing – review & editing.

## Competing interests

The authors declare no competing interests.
