## [Transparent Peer Review file · Communications Psychology]

Non-random patterns in the co-occurrence and accumulation of adverse life events in two national panel datasets

Corresponding Author: Ms Kyra Evers

Version 0:

Decision Letter:

Dear Ms Evers,

Thank you for your patience during the peer-review process. Your manuscript titled "Non-random patterns in the co-occurrence and accumulation of adverse life events in two national panel datasets" has now been seen by 2 reviewers, and I include their comments at the end of this message. They find your work of interest but raised some important points. We are interested in the possibility of publishing your study in *Communications Psychology*, but would like to consider your responses to these concerns and assess a revised manuscript before we make a final decision on publication.

We therefore invite you to revise and resubmit your manuscript, along with a point-by-point response to the reviewers. Please highlight all changes in the manuscript text file.

The reviewers identify a critical gap in the manuscript's failure to incorporate genetically-informed research, particularly co-twin control designs that demonstrate ALE-mental health associations remain significant even after controlling for shared genetic factors, which would strengthen the authors' causal arguments about self-reinforcing ALE patterns. Both reviewers raise methodological concerns including potential collider bias when controlling for all other ALEs simultaneously, unclear model comparison procedures for non-nested models, and possible selection bias from restricting analyses to participants with complete 20-year data. The reviewers emphasize the need for clearer conceptual distinction between self-reinforcing ALE accumulation versus genetic and environmental confounding factors, suggesting the introduction should better contrast these competing explanations for why adversities cluster over time. Please also include a description in the Methods section regarding ethics approval, or its waiver, for your secondary data analysis.

I am attaching an Editorial Requests Table that details critical reporting requirements for the revised manuscript. Please attend to each item and ensure your manuscript is fully compliant. If your revised manuscript is not aligned with these requests on major issues, such as those concerning statistics, it may be returned to you for further revisions without re-review.

Please submit the following items:

- Revised manuscript
- Point-by-point response to the referees' comments
- Cover letter (as a separate document)
- <https://www.nature.com/documents/nr-reporting-summary.pdf> Nature Research Reporting Summary
- Completed Editorial Request Table (attached).

via this link: Link Redacted .

Additional guidance is available in our style and formatting guide Communications Psychology formatting guide.

Best regards,

Anna-Lena Schubert

Anna-Lena Schubert, PhD
Editorial Board Member
Communications Psychology
orcid.org/0000-0001-7248-0662

REVIEWER EXPERTISE:

Reviewer #1: Stress and adversity / life course epidemiology / longitudinal modeling

Reviewer #2: Stress and adversity / life course epidemiology / longitudinal modeling

REVIEWER REPORTS:

Reviewer #1 (Remarks to the Author):

Thank you for the opportunity to review this interesting manuscript on patterns of occurrences of adverse life events (ALEs). The study is well executed and well written. I hope the authors find my comments useful in further improving the manuscript.

Major Points

Introduction

Page 2: The authors draw in part upon genetically informed studies to shed light on patterns of ALE exposure and how these events are linked to mental health. However, an important piece is missing. The authors describe shared genetic factors and the potential for correlated pathways, but do not mention the many co-twin control designs that have found that associations between ALE exposure and mental health, while attenuated within twin pairs, nonetheless remain modest-to-considerable (e.g., Kendler et al., 1999; Bjørndal et al., 2023; Baldwin et al., 2023).

I recommend that the authors review these genetically informed designs, which by virtue of accounting for unmeasured shared confounding offer significant advantages compared to other observational designs. This may also help support some of the points made using causal language regarding the relationship between ALE exposure and outcomes.

Page 2: The description of gene environment correlation (rGE) is important. However, the direct correspondence between rGE and stress generation is unclear. rGE may act through many pathways beyond stress generation alone. I recommend that the authors revise this section for clarity. rGE processes are well-known in the behavior genetics literature (and the authors include some relevant references) and may deserve a dedicated paragraph or at least a more elaborated description. This discussion should also be more clearly separated from stress generation. Some recent UK BioBank work also addresses rGE in stress exposure and mental health, which the authors may find relevant.

Pages 2 to 3: I commend the authors for their excellent writing, especially in the description of stress proliferation. However, this paragraph would benefit from a bit more emphasis on how ALE exposure may influence internal characteristics or

resources (for example, coping strategies, as noted). Might it not be the case interpersonal ALEs affect sustained self perceptions or beliefs (for example, a break-up may lead someone to believe they cannot succeed in romantic relationships), which could increase the likelihood of future ALEs (e.g., further break-ups)?

Methods

Can the authors elaborate on how the models were compared, given that only AIC and BIC (to my understanding) are listed as fit indices? It is noted that "Fit comparisons subtract the fit of the model on the right from that of the model on the left," but it is not clear how, for example, the fit of the Polya urn model can be compared using the same metrics as the alternative models. These models are not nested, correct? If so, could the authors report fit metrics that would better allow readers to evaluate model differences?

Additionally, can the authors clarify whether they compared only models in which ALE occurrences were either (a) entirely random (Poisson) or (b) entirely self sustaining (Polya urn)? While possibly beyond the scope of this paper, it is worth noting that both extremes may be less realistic than a model in which some ALEs occur randomly and others reflect self sustaining feedback (that is, an intermediate model). If these models include these attributes, it could strengthen the rationale if the authors mention this.

Discussion

I found the first summary paragraph somewhat repetitive, given the clear summaries provided earlier. I suggest shortening this or earlier summary paragraphs to improve clarity.

The authors rightly mention that associations may in part be influenced by recall bias (for example, due to current emotional distress). However, the role of such bias in shaping the study's findings, particularly the observed patterns of co-occurrences, is not clearly discussed. It seems plausible that mental health consequences of ALE exposure (for example, depression) could affect how individuals report ALEs the following year (lag one). I encourage the authors to elaborate on this.

Discussion: Conceptualisation

While I agreed with many of the authors' points, I noted again the omission of some important literature, especially from genetically informed research. This work has demonstrated for decades that ALEs are heritable, described as "pervasive in extent and modest to moderate in impact" (Kendler and Baker, 2007, *Psychological Medicine*). Consequently, in my view, most researchers (at least in genetically informative ALE/SLE research) places less emphasis on the independent versus dependent ALE distinction than may be implied in the manuscript as is.

In relation to this, it is well established that associations between dependent ALEs and mental health outcomes are substantially confounded (Kendler and Gardner, 2010, *JAMA Psychiatry*). The manuscript does not fully address how genetic or other unmeasured confounding might influence the results. This is a key consideration that should be acknowledged. It does seem likely that much of the same genetic confounding (e.g., mediated by personality traits) could be involved in the lag-1 associations.

Minor Points

Abstract

The meaning of "structural predictors" of ALEs was unclear to me and is likely to be unclear to other readers. It may refer to predictors of patterns or occurrences of ALEs, but I recommend the authors revise this for clarity. Similarly, "structural patterns of ALEs" is slightly unclear in the abstract. This phrase could be interpreted as suggesting a factor analytic approach or an examination of latent structure.

Introduction, first paragraph

This is a well written paragraph that offers a broad overview of the impact of ALEs. However, I recommend greater caution in the use of causal language (for example, "The effects of ALEs propagate"), given the possibility of unmeasured confounding.

Line 203: Typo in "strongest."

Line 214: There are two consecutive periods ("..").

Reviewer #2 (Remarks to the Author):

This study investigates how different adverse life events (ALEs) are associated with each other, and accumulate across time, in two large representative samples. The authors find that many ALEs are associated with each other, both contemporaneously and across time. In addition, they show that the accumulation of ALEs over time is best explained by a model assuming identical initial risk and a self-reinforcing increase over time.

I appreciated the opportunity to review this manuscript. It has multiple noteworthy strengths, including its thoughtful framing and embedding in previous literature, the overall clarity of argument and writing, and the meticulous methods used. It offers an important contribution to the field, opening up exciting new directions for future research.

1. In the introduction, the authors discuss different reasons for why adversity exposures could become clustered over time, including personal and environmental predictors. The spreading effects of events themselves is posed as one such reason,

alongside personal characteristics, genetic factors, and structural (environmental) factors. Based on this framing, I was anticipating a more direct contrasting between (some of) these factors (e.g., is clustering between exposures the result of (self)-reinforcement or of confounding factors?). A (slight) rewriting of the latter half of the introduction might help to better set up a contrast between these explanations, and make more explicit why it is worthwhile to investigate (self)-reinforcing effects in addition to often studied predictors of ALEs.

2. The co-occurrence and autocorrelation models control for all other ALEs. This approach risks the influence of potential colliders. How much do the results change when excluding the other ALEs as control variables, relative to the models including them? Are there any statistical and/or conceptual reasons to suspect that colliders may or may not have been an issue?

3. The authors investigated the accumulation of event counts in a restricted sample of people who had continuous data across twenty years. It would be good to provide some information on how these subsamples compare to the complete samples in terms of demographic/SES indicators. I can imagine that people who did not miss a single assessment in twenty years are generally better off, which could potentially bias the model results a bit. If the authors think it is worthwhile, they may consider adding robustness checks to the supplemental materials based on other approaches of handling missing data.

4. I was left wondering what the implication is of the large explained variance of random effects relative to fixed effects. The authors may consider discussing this in a bit more detail in the discussion, for instance, by linking it back to the potential predictors of ALEs discussed in the introduction.

5. In most panels of Figure 4, it is difficult/impossible to discern the lines of the Frailty and Polya urn models. This is mostly due to the cluttering effect of the empirical data, with its thick red dots and vertical lines. Perhaps the authors could try out alternative ways of plotting that would better show the nuances of model fit between these models.

Stefan Vermeent

Version 1:

Decision Letter:

Dear Ms Evers,

Your manuscript titled "Non-random patterns in the co-occurrence and accumulation of adverse life events in two national panel datasets" has now been seen by our reviewers, whose comments appear below. In light of their advice I am delighted to say that we are happy, in principle, to publish a suitably revised version in Communications Psychology.

We therefore invite you to revise your paper one last time to address the remaining concerns of our reviewers and a list of

editorial requests. At the same time we ask that you edit your manuscript to comply with our format requirements and to maximise the accessibility and therefore the impact of your work.

EDITORIAL REQUESTS:

SUBMISSION INFORMATION:

In order to accept your paper, we require the files listed here <https://www.nature.com/documents/commsj-file-checklist.pdf> .

OPEN ACCESS:

* DATA AVAILABILITY:

Link Redacted

Best regards,

Jennifer Bellingtier

Jennifer Bellingtier, PhD
Senior Editor
Communications Psychology

Anna-Lena Schubert, PhD
Editorial Board Member
Communications Psychology
orcid.org/0000-0001-7248-0662

REVIEWER EXPERTISE:

Reviewer #1: Stress and adversity / life course epidemiology / longitudinal modeling

Reviewer #2: Stress and adversity / life course epidemiology / longitudinal modeling

REVIEWERS' COMMENTS:

Reviewer #1 (Remarks to the Author):

I thank the authors for their revisions and wish to congratulate them on this work. A few minor comments:

- I believe Ref 10 in the revised version is not a genetically informative design and therefore does not support the point regarding co-twin control designs.
- P. 4: "... based on the list developed by 78" - you may wish to rewrite this slightly considering the numbered reference style.
- P. 4: "No sampling weights were applied in order to make the results more interpretable". It may not be immediately clear to the reader why applying sample weights would reduce interpretability.

Reviewer #2 (Remarks to the Author):

I acted as reviewer for the previous version of this manuscript.

I appreciate the author's responsiveness to my comments. The additional (sensitivity) analyses address all my prior concerns. I have no further comments on the revised manuscript.

November 9, 2025

Dear Dr. Schubert,

Thank you for giving us the opportunity to submit a revised version of our manuscript “Non-random patterns in the co-occurrence and accumulation of adverse life events in two national panel datasets” for publication in *Communications Psychology*. We greatly appreciate the time and effort that you and the reviewers dedicated to providing such constructive and thoughtful feedback. We have highlighted the changes within the manuscript, and address each point below.

Kind regards,

Kyra Evers
Denny Borsboom
Eiko Fried
Fred Hasselman
František Bartoš
Lourens Waldorp

Reviewer 1:

1. *Introduction*

- (a) *Page 2: The authors draw in part upon genetically informed studies to shed light on patterns of ALE exposure and how these events are linked to mental health. However, an important piece is missing. The authors describe shared genetic factors and the potential for correlated pathways, but do not mention the many co-twin control designs that have found that associations between ALE exposure and mental health, while attenuated within twin pairs, nonetheless remain modest-to-considerable (e.g., Kendler et al., 1999; Bjørndal et al., 2023; Baldwin et al., 2023).*

I recommend that the authors review these genetically informed designs, which by virtue of accounting for unmeasured shared confounding offer significant advantages compared to other observational designs. This may also help support some of the points made using causal language regarding the relationship between ALE exposure and outcomes.

We would like to thank the reviewer for this helpful suggestion. We have added the following to the relevant paragraph in the Introduction section:

“Importantly, evidence from genetically informed co-twin control designs indicates that, although associations between adverse life event exposure and mental health outcomes are attenuated within twin pairs, they nonetheless remain modest to considerable in magnitude, suggesting that these links cannot be fully explained by shared genetic or familial confounding^{9;10;11}.”

- (b) *Page 2: The description of gene environment correlation (rGE) is important. However, the direct correspondence between rGE and stress generation is unclear. rGE may act through many pathways beyond stress generation alone. I recommend that the authors revise this section for clarity. rGE processes are well-known in the behavior genetics literature (and the authors include some relevant references) and may deserve a dedicated paragraph or at least a more elaborated description. This discussion should also be more clearly separated from stress generation. Some recent UK BioBank work also addresses rGE in stress exposure and mental health, which the authors may find relevant.*

We agree that the relation between stress generation and different types of rGE was unclearly formulated. Our intention with discussing the stress generation framework was to provide more insight into why dependent events occur from the perspective of personality and clinical science. We have revised this paragraph to better distinguish between stress generation and the different types of rGE, by first briefly describing passive, evocative, and active gene-environment correlations, and then introducing stress generation as a separate framework:

“Passive gene-environment correlation occurs when parents transmit both genetic propensities and correlated environments to their offspring; evocative gene-environment correlation occurs when genetically influenced traits elicit reactions from others; and active gene-environment correlation occurs when individuals select or create environments aligned with their genetic propensities. Research from clinical and personality science elaborates on the mechanisms by which personality, psychopathological, and behavioural characteristics can contribute to dependent life events, a phenomenon termed *stress generation*^{31;32;33;34}. The stress generation framework identifies four mechanisms^{32;36;37} [...]. These mechanisms illustrate how genetically influenced personality traits and behaviours can increase exposure to dependent adverse life events through environmental shaping.”

- (c) *Pages 2 to 3: I commend the authors for their excellent writing, especially in the description of stress proliferation. However, this paragraph would benefit from a bit more emphasis on how ALE exposure may influence internal characteristics or resources (for example, coping strategies, as noted). Might it not be the case interpersonal ALEs affect sustained self perceptions or beliefs (for example, a break-up may lead someone to believe they cannot succeed in romantic relationships), which could increase the likelihood of future ALEs (e.g., further break-ups)?*

We would like to thank the reviewer for their compliments. We agree that more attention to the psychological resources is worthwhile. We have added the following to the paragraph on stress proliferation:

“Importantly, resources are not static, but may in turn be affected by adverse life events. On the one hand, experiencing adversity can help to build resilience to future adversity^{2;61;59;62;63}, also known as post-traumatic growth^{64;65}. On the other hand, adverse life events have been linked to the onset of mental illness^{9;11;66;67}, diminished self-concept and self-esteem^{68;69;70}, and shattered core assumptions about safety and control^{71;72}.”

2. Methods

- (a) *Can the authors elaborate on how the models were compared, given that only AIC and BIC (to my understanding) are listed as fit indices? It is noted that “Fit comparisons subtract the fit of the model on the right from that of the model on the left,” but it is not clear how, for example, the fit of the Polya urn model can be compared using the same metrics as the alternative models. These models are not nested, correct? If so, could the authors report fit metrics that would better allow readers to evaluate model differences?*

The models are indeed non-nested, but both the AIC and BIC are appropriate for non-nested model comparisons^{94;95}, with lower values indicating better fit. We have added this in the Methods section for clarity:

“In the accumulation analysis, models were compared using Akaike Information Criterion (AIC) and Bayesian Information Criterion (BIC). Both metrics are appropriate for non-nested model comparisons^{94;95}. AIC emphasizes predictive accuracy while BIC imposes stronger penalties for model complexity. For clarity, we report fit comparisons as $\Delta\text{AIC} = \text{AIC}(\text{focal model}) - \text{AIC}(\text{comparison model})$ and similarly for BIC; a positive difference indicates the comparison model fits better.”

- (b) *Additionally, can the authors clarify whether they compared only models in which ALE occurrences were either (a) entirely random (Poisson) or (b) entirely self sustaining (Polya urn)? While possibly beyond the scope of this paper, it is worth noting that both extremes may be less realistic than a model in which some ALEs occur randomly and others reflect self sustaining feedback (that is, an intermediate model). If these models include these attributes, it could strengthen the rationale if the authors mention this.*

We agree that the relation between these accumulation models could be more clearly described. We have clarified in the manuscript that the Polya urn model combines both inter-individual differences in baseline risk, randomness over time, and self-reinforcement. In the Methods section, we have expanded the description of the differences between these models, emphasizing that all three models include a degree of randomness:

“A Poisson, frailty, and Polya urn model thus offer three distinct causal mechanisms underlying the accumulation of adverse life events. In a Poisson model, events are independent over time and across the population, as risk is static and homogeneous. In a frailty model, risk is differentially distributed across the population, and is *independent* of the events which already occurred conditioned on the combination of individual and household risk. In a Polya urn model with random effects, in addition to a heterogeneous distribution of risk across the population, the events *themselves* increase the probability of future adverse life events. In other words, the Poisson and frailty model assume static risk, whereas risk in the Polya urn model is history-dependent and dynamic. Importantly, the self-reinforcement process in the Polya urn model is not deterministic, because all models involve a degree of randomness as they sample events from a distribution given a certain probability.”

In the Results section, we partially repeat these differences:

“As a random distribution process of events fits the data poorly, we considered two alternative stochastic processes which do include dependencies: a frailty model, in which risk is heterogeneously distributed across the population but random across time, and a Polya urn model, in which baseline risk is also heterogeneously distributed across the population, but the risk of events is self-reinforcing over time”

In the Discussion section, we have better described that the Polya urn model with random effects includes inter-individual differences in baseline risk:

“The accumulation analysis indicates that the risk of adverse life events is best modelled as heterogeneously distributed across the population with an additional self-reinforcement process, such that the risk of adverse life events depends on one’s history of adversity.”

3. Discussion

- (a) *I found the first summary paragraph somewhat repetitive, given the clear summaries provided earlier. I suggest shortening this or earlier summary paragraphs to improve clarity.*

We agree that the summary of the results in the first paragraph of the Discussion is too repetitive. To still give readers a bird’s eye view of the results, we included interim summaries in the Results section, removing them from the Discussion.

- (b) *The authors rightly mention that associations may in part be influenced by recall bias (for example, due to current emotional distress). However, the role of such bias in shaping the study’s findings, particularly the observed patterns of co-occurrences, is not clearly discussed. It seems plausible that mental health consequences of ALE exposure (for example, depression) could affect how individuals report ALEs the following year (lag one). I encourage the authors to elaborate on this.*

To more specifically discuss the potential of mental health consequences on adverse life event reporting, we have included the following in the Discussion:

“For instance, adverse life events can precipitate mental health problems such as depression and anxiety^{99;66;67}, which in turn may influence subsequent event reporting through mechanisms such as heightened negative memory bias increased rumination on events^{100;101}. This creates potential for spurious lag-1 associations wherein mental health consequences of events in one year inflate the reporting of events in the subsequent year, independent of actual event occurrence.”

- (c) *While I agreed with many of the authors’ points, I noted again the omission of some important literature, especially from genetically informed research. This work has demonstrated for decades that ALEs are heritable, described as “pervasive in extent and modest to moderate in impact” (Kendler and Baker, 2007, Psychological Medicine). Consequently, in my view, most researchers (at least in genetically informative ALE/SLE research) places less emphasis on the independent versus dependent ALE distinction than may be implied in the manuscript as is.*

In relation to this, it is well established that associations between dependent ALEs and mental health outcomes are substantially confounded (Kendler and Gardner, 2010, JAMA Psychiatry). The manuscript does not fully address how genetic or other unmeasured confounding might influence the results. This is a key consideration that should be acknowledged. It does seem likely that

much of the same genetic confounding (e.g., mediated by personality traits) could be involved in the lag-1 associations.

We agree that we have insufficiently highlighted that our findings cannot be interpreted as causal and that the potential of confounders should be emphasized. To remedy this, we have added a new paragraph in the Discussion:

“For all analyses, differences between individuals and households were stronger predictors of event occurrence than concurrent or prior adverse life events. That is, individual- and household-level characteristics as captured by random effects accounted for a much larger part of the explained variance than fixed effects. This aligns with genetically informed research demonstrating substantial genetic influence on adverse life event exposure and reporting²⁹, described as “pervasive in extent and modest to moderate in impact”²², with genetic factors influencing exposure through personality traits, behavioural tendencies, and other stable characteristics. Such variables are captured by the random intercepts in our models, which model unobserved variables that elevate overall adverse life event risk. However, the random intercepts cannot account for unmeasured characteristics that may differentially influence specific types of event sequences beyond general elevation in risk. For instance, while impulsivity may increase baseline event rates generally (captured by the random intercept), it might also specifically increase both financial problems and subsequent relationship conflicts more than other event pairs, potentially creating spurious lag-1 associations between these particular events. Genetically informed designs, such as co-twin control studies comparing associations between versus within twin pairs, could help disentangle trait-mediated associations from genuine event-to-event effects²⁹. Another important confounder to consider are events themselves, as individuals experiencing the most events may be more likely to drop out of the study. Selective attrition would be expected to underestimate self-reinforcing effects of adverse life events. Given these limitations, our findings are best interpreted as descriptive patterns, rather than reflecting direct causal effects between events.”

4. *Minor Points*

- (a) *The meaning of “structural predictors” of ALEs was unclear to me and is likely to be unclear to other readers. It may refer to predictors of patterns or occurrences of ALEs, but I recommend the authors revise this for clarity. Similarly, “structural patterns of ALEs” is slightly unclear in the abstract. This phrase could be interpreted as suggesting a factor analytic approach or an examination of latent structure.*

We would like to thank the reviewer for pointing this out. To improve clarity, we have replaced “structural predictors” with “risk factors” and “structural patterns” with “non-random patterns” in the abstract.

- (b) *This is a well written paragraph that offers a broad overview of the impact of ALEs. However, I recommend greater caution in the use of causal language (for example, “The effects of ALEs propagate”), given the possibility of unmeasured confounding.*

To refrain from causal language, we have reformulated the following sentence in the Introduction: “The effects of adverse life events can extend across multiple domains of life¹⁴, potentially affecting not only physical and mental well-being, but also developmental, financial, occupational, and social health.”

- (c) *Line 203: Typo in “strongest.”*

This is now corrected.

- (d) *Line 214: There are two consecutive periods (“..”).*

This is now corrected.

Reviewer 2:

1. *In the introduction, the authors discuss different reasons for why adversity exposures could become clustered over time, including personal and environmental predictors. The spreading effects of events*

themselves is posed as one such reason, alongside personal characteristics, genetic factors, and structural (environmental) factors. Based on this framing, I was anticipating a more direct contrasting between (some of) these factors (e.g., is clustering between exposures the result of (self)-reinforcement or of confounding factors?). A (slight) rewriting of the latter half of the introduction might help to better set up a contrast between these explanations, and make more explicit why it is worthwhile to investigate (self)-reinforcing effects in addition to often studied predictors of ALEs.

We would like to thank the reviewer for helping to improve the framing of our study. We have revised the second to last paragraph in the Introduction to better relate our study to the above discussed common causes and self-reinforcing pathways, elaborating on our research questions:

“In summary, exposure to adverse life events is distinctly non-random, involving a complex web of numerous common causes and self-reinforcing pathways. Over time, their interaction has the potential to culminate in clusters of adversity and large individual differences. Whereas most research has focused on identifying risk and resilience factors of adverse life events or quantifying their impact, few studies examine patterns in events themselves^{10;73}. Given the multivariate and interacting nature of these processes – personality, psychopathological, and behavioural characteristics, socio-economic factors, and stress proliferation – there are compelling reasons to expect structural patterns in how adverse life events cluster and accumulate. However, the specific patterns that emerge from these processes are not straightforward to predict. Which types of events tend to co-occur? Which events predict which subsequent events? How do adverse life events accumulate over decades?”

2. *The co-occurrence and autocorrelation models control for all other ALEs. This approach risks the influence of potential colliders. How much do the results change when excluding the other ALEs as control variables, relative to the models including them? Are there any statistical and/or conceptual reasons to suspect that colliders may or may not have been an issue?*

The risk of colliders influencing the results is indeed a valid concern. In the case of lag-1 associations, the risk of colliders is less of a problem, as it is not possible for an event to be an outcome of an event in the next year. However, it is regardless good to assess the impact of adjusting for other events. We have added a sensitivity analysis, in which we reran all models without adjusting for other events.

We describe this in the Methods section:

“Both the contemporaneous and lag-1 co-occurrence analysis adjust for all other events, which has the potential to distort associations, such as when controlling for mediators or colliders⁷⁹. For instance, a major worsening in finances can be an outcome of both being fired and separation from spouse, such that controlling for a major worsening in finances (a collider) can misrepresent the relation between being fired and separation from spouse. To assess the impact of this modelling choice, we compared our fully adjusted models (controlling for all other events) to bivariate models that included only one predictor event at a time, while retaining the same random effects structure for individuals and households and fitting to the same data. To compare associations, we calculated the percentage difference between unadjusted and adjusted odds ratios as: $100 \times (OR_{unadjusted} - OR_{adjusted}) / OR_{adjusted}$. In addition, we identified which significant adjusted associations reversed direction when not adjusting for other events (i.e., $OR_{unadjusted} > 1$ and $OR_{adjusted} < 1$, or vice versa).”

The findings are reported in the Results section:

“The sensitivity analysis comparing adjusted and unadjusted odds ratios showed that all significant associations maintained the same direction, with unadjusted estimates being larger than adjusted estimates (SHP: mean = 4.61%, median = 2.80%, SD = 6.58; HILDA: mean = 43.16%, median = 29.38%, SD = 38.69; Supplementary Figs. S5-S6). This pattern suggests that controlling for other events primarily reduced the magnitude of associations, but did not change their direction.”

“The sensitivity analysis for lag-1 associations showed similar patterns as the contemporaneous analysis, with unadjusted estimates being larger than adjusted estimates (SHP: mean = 1.27%, median = 1.24%, SD = 11.37; HILDA: mean = 10.40%, median = 7.08%, SD = 14.43; Supplementary Figs. S8-S9). All significant adjusted associations maintained the same direction when unadjusted for other events, except one: the death of a spouse or child in HILDA, which changed from a negative adjusted association (OR = 0.62 [0.49, 0.79]) to a positive unadjusted association (OR = 1.55 [1.27, 1.89]).”

3. *The authors investigated the accumulation of event counts in a restricted sample of people who had continuous data across twenty years. It would be good to provide some information on how these subsamples compare to the complete samples in terms of demographic/SES indicators. I can imagine that people who did not miss a single assessment in twenty years are generally better off, which could potentially bias the model results a bit. If the authors think it is worthwhile, they may consider adding robustness checks to the supplemental materials based on other approaches of handling missing data.*

We understand the concern that a sample without attrition across 20 years is plausibly different from the complete sample. We chose not to impute the data as this can introduce additional bias when the mechanisms driving missingness (attrition due to severe adverse events) are complex and not missing at random. As a robustness check, we repeated the accumulation analysis for 10 and 15 consecutively observed years (Supplementary Tables S5 and S6), which showed the same pattern of findings: the frailty model fit better than the Poisson model, and the Polya urn model fit better than the frailty model. This robustness check supports that our findings do not only hold for a sample which is observed across 20 consecutive years.

We have included more information on the subsamples in Supplementary Figs. S1-S2, including age, sex, socio-economic status (HILDA) and number of years of education and working status (SHP), as the same variables were not available for both datasets. The demographics across the subsamples remain largely the same, except for age, which converges to a normal distribution with an increasing consecutive observation period.

4. *I was left wondering what the implication is of the large explained variance of random effects relative to fixed effects. The authors may consider discussing this in a bit more detail in the discussion, for instance, by linking it back to the potential predictors of ALEs discussed in the introduction.*

We have dedicated a separate paragraph in the Discussion to discussing these in light of potential confounders:

“For all analyses, differences between individuals and households were stronger predictors of event occurrence than concurrent or prior adverse life events. That is, individual- and household-level characteristics as captured by random effects accounted for a much larger part of the explained variance than fixed effects. This aligns with genetically informed research demonstrating substantial genetic influence on adverse life event exposure and reporting²⁹, described as “pervasive in extent and modest to moderate in impact”²², with genetic factors influencing exposure through personality traits, behavioural tendencies, and other stable characteristics. Such variables are captured by the random intercepts in our models, which model unobserved variables that elevate overall adverse life event risk. However, the random intercepts cannot account for unmeasured characteristics that may differentially influence specific types of event sequences beyond general elevation in risk. For instance, while impulsivity may increase baseline event rates generally (captured by the random intercept), it might also specifically increase both financial problems and subsequent relationship conflicts more than other event pairs, potentially creating spurious lag-1 associations between these particular events. Genetically informed designs, such as co-twin control studies comparing associations between versus within twin pairs, could help disentangle trait-mediated associations from genuine event-to-event effects²⁹. Another important confounder to consider are events themselves, as individuals experiencing the most events may be more likely to drop out of the study. Selective attrition would be expected to underestimate self-reinforcing effects of adverse life events. Given these limitations, our findings are best interpreted as descriptive patterns, rather than reflecting direct causal effects between events.”

5. *In most panels of Figure 4, it is difficult/impossible to discern the lines of the Frailty and Polya urn models. This is mostly due to the cluttering effect of the empirical data, with its thick red dots and vertical lines. Perhaps the authors could try out alternative ways of plotting that would better show the nuances of model fit between these models.*

The visual difference between the frailty and Polya urn model is indeed hard to detect in Figure 4. The difference between these two models becomes most apparent when plotting the full range of the predicted cumulative number of events, where the frailty model predicts much longer tails than the Polya urn model. In Figure 4, we have clipped the distributions to the observed range of the empirical

data, as plotting the full range obscures the fit in the region where most observations fall and makes visual comparison to the empirical distribution difficult. To show the predicted range of all models across their full distribution, we have included Supplementary Fig. S19. In the revised version of Figure 4, we have decreased the size of the empirical dots to improve visibility of the model predictions, but the difference between the frailty and Polya urn model is best seen in the new Supplementary Figure.